# Design, Synthesis and Biological Evaluation of Novel 1,3,5-Triazines: Effect of Aromatic Ring Decoration on Affinity to 5-HT_7_ Receptor

**DOI:** 10.3390/ijms232113308

**Published:** 2022-11-01

**Authors:** Damian Kułaga, Anna Karolina Drabczyk, Grzegorz Satała, Gniewomir Latacz, Anna Boguszewska-Czubara, Damian Plażuk, Jolanta Jaśkowska

**Affiliations:** 1Department of Organic Chemistry and Technology, Faculty of Chemical Engineering and Technology, Cracow University of Technology, ul. Warszawska 24, 31-155 Kraków, Poland; 2Department of Medicinal Chemistry, Maj Institute of Pharmacology, Polish Academy of Sciences, ul. Smętna 12, 31-343 Kraków, Poland; 3Department of Technology and Biotechnology of Drugs, Jagiellonian University Medical College, ul. Medyczna 9, 30-688 Kraków, Poland; 4Department of Medical Chemistry, Medical University of Lublin, ul. Chodźki 4a, 20-093 Lublin, Poland; 5Laboratory of Molecular Spectroscopy, Department of Organic Chemistry, Faculty of Chemistry, University of Lodz, ul. Tamka 12, 91-403 Łódź, Poland

**Keywords:** serotonin, microwave synthesis, CNS

## Abstract

Considering the key functions of the 5-HT_7_ receptor, especially in psychiatry, and the fact that effective and selective 5-HT_7_ receptor ligands are yet to be available, in this work, we designed and synthesized novel 1,3,5-triazine derivatives particularly based on the evaluation of the effect of substituents at aromatic rings on biological activity. The tested compounds showed high affinity to the 5-HT_7_ receptor, particularly ligands N^2^-(2-(5-fluoro-1H-indol-3-yl)ethyl)-N^4^-phenethyl-1,3,5-triazine-2,4,6-triamine **2** (*K*_i_ = 8 nM) and N^2^-(2-(1H-indol-3-yl)ethyl)-N^4^-(2-((4-fluorophenyl)amino)ethyl)-1,3,5-triazine-2,4,6-triamine **12** (*K*_i_ = 18 nM) which showed moderate metabolic stability, and affinity to the CYP3A4 isoenzyme. As for the hepatotoxicity evaluation, the tested compounds showed moderate cytotoxicity only at concentrations above 50 µM. Compound **12** exhibited less cardiotoxic effect than **2** on Danio rerio in vivo model.

## 1. Introduction

The 5-HT_7_ receptor (5-HT_7_R) is one of seven types of protein G-coupled aminergic serotonin receptors [1]. It is found in the central nervous system, mainly in the brain, in which it has regulatory functions, for example (the day–night cycle), and may affect behavior, mood, emotions, or memory [2,3]. The receptor is also expressed outside the central nervous system, e.g., in the intestines [4], mammary glands [5], lungs [6], or prostate [7].

Three human splicing variants (h5-HT_7(a)_, h5HT_7(b)_, and h5-HT_7(d)_) have been reported so far. As long as the h5-HT_7(a)_ and h5-HT_7(b)_ variants do not differ significantly in terms of the number of amino acids in the C-terminal tail, for example, and maintain similar pharmacological properties, the h5-HT_7(d)_ variant shows the greatest differences in the C-terminal tail, which may lead to slightly different functionality [1]. The primary signaling pathway for the 5-HT_7_ receptor involves the receptor binding the ligand, followed by phosphorylation of subunit G combined with its dissociation into subunit G_S_ and heterodimer G_12_. In the subsequent stage, protein G_S_ (canonical signaling) is activated, which triggers isoform CA (adenyl cyclase) and leads to an intrinsic increase in cAMP (cyclic adenosine monophosphate) levels. cAMP induces PKA (protein kinase A) expression, which in turn induces further phosphorylation of other proteins, for example, on Ras, ERK, and Akt pathways. It has also been shown that protein Gs and protein G_12_ (non-canonical signaling) are activated by 5-HT_7_R, which leads to further signaling in the cell and results in neurite outgrowth, synaptogenesis, and neuronal excitability [8,9].

It is well-known that 5-HT_7_R is a binding site for several bioactive compounds with antidepressive or anxiolytic effects [2,3]. However, as the ligand–receptor complex may indirectly affect downstream signalization, 5-HT_7_ receptor ligands may affect the expression of respective kinases or proteins (whose overexpression occurs in tumor cells). It is owing to this property, among other things, that some 5-HT_7_ receptor antagonists or agonists have anticancer [10,11,12] or anti-inflammatory effects [13,14]. Based on the significant functions of 5-HT_7_R and also the therapeutic effects that can be achieved owing to this receptor, it is justified to discover novel, selective and safe ligands of the receptor.

We recently proved that it was possible to forgo the well-known arylpiperazine pharmacophore while maintaining activity and selectivity toward 5-HT_7_R [15]. The lead motif in the studies so far was an unsubstituted tryptamine core connected to aminotriazine. However, literature reports are available in which the incorporation of substituents at the indole C-5 position resulted in increased affinity toward 5-HT_7_R. A particular effect was noted, with substituents being electron withdrawal groups (EWGs) [16,17]. Therefore, we decided in this report to synthesize a group of compounds 1–6 (Figure 1, type 1) with a substituted indole C-5 position (by EWGs and EDGs (electron donating groups)) and subsequently to evaluate their effect on affinity toward the 5-HT_7_ receptor. Another argument for synthesizing indole derivatives is the effect of heteroatoms on ADME-T parameters, particularly the fluorine atom [18,19]. We showed [20] that the compounds tested without any ring substituent (Figure 1A) had low in vitro metabolic stability. By incorporating chlorine in the structure [15] (Figure 1B), metabolic stability increased with a slight decrease in cytotoxicity in the HepG2 cell line. This correlation is also confirmed by the original report by Mattson et al. [21], in which the incorporation of fluorine atoms in the molecule increased stability almost five times compared to the unsubstituted compound (Figure 2).

In our previous paper [15], we also investigated the effect of substituents (*ortho*-OMe and 2,3-Cl_2_) at the phenyl ring attached to the aminoethyl chain on affinity toward 5-HT_7_R. The substituents were found to reduce receptor affinity, but the effect of a specific substituent position on activity was not tested. To investigate this aspect, we decided to synthesize a group of ligands **7**–**15** (Figure 1, type 2) containing Cl, F, and OMe at positions *ortho*-, *meta*-, and *para*- and to evaluate their effects on affinity toward 5-HT_7_R. We also decided to incorporate more complex substituents into the structures of studied compounds (phthalimide **16** and benzimidazole **17**–**19** cores, Figure 1, type 3) due to the potential formation of hydrogen bonds to stabilize the ligand–receptor additionally. Unexpectedly, we prepared compounds **20**–**22** without the aminoethyl chain in our synthesis experiments (Figure 1, type 4). To determine the effect of the lack of an alkyl linker between the triazine and the aromatic system on affinity to the receptors in question, we also evaluated these compounds in in vitro tests. All the resulting compounds were tested in an extended receptor panel, including affinity toward 5-HT_1A_, 5-HT_2A_, 5-HT_6_, and D_2_ receptors, to determine the selectivity of the compounds. Bioconformation and key interactions involved in the forming the ligand–receptor complex were proposed for active structures. The two best compounds were evaluated in terms of safety and bioavailability in in vitro ADME-Tox tests.

## 2. Results

### 2.1. Chemistry

Final type 1 compounds were synthesized according to Figure 2, starting from commercially available 5-substituted indoles **23–28**, converted to 3-substituted aldehydes **29–34** via Vilsmeier-Haack formylation with 90–100% yields. In a subsequent stage, the aldehydes were subjected to the Henry reaction to obtain nitrovinyl derivatives **35–40**. The synthesis of the derivatives initially followed patent [22] under conventional reflux of the reaction mixture. As long as yields of more than 90% were obtained in a small scale of 0.5–1 g, side products formed in the reaction mixture (according to TLC) with low yields of 40–56% when the scale was increased to 5 g. However, we found that performing the reactions under microwave irradiation (P = 85 W) for 20 min allows obtaining desired products in more than 90% yields, irrespective of the scale. Crude **35–40** were used in the next stage without any further purification. As for compound **35**, another two stages involved reduction: first of the double bond using a mild reducing agent (NaBH_4_) to give **41** and subsequently reduction of the nitro group with zinc in boiling 36% HCl solution, to finally give compound **47**. As compounds **36–40** lacked the nitrile substituent, which could also be reduced in harsher conditions, simultaneous reduction of the double bond and the nitro group could be performed in the presence of LiAlH_4_ to give final 5-substituted tryptamines **42–46**, respectively. The subsequent stage was the synthesis of core compounds **51**. Cyanuric chloride **48** was reacted with phenylethylamine **49** at 0 °C, and the resulting product **50** was treated with ammonia water. Synthesis was conducted at room temperature for 5 h, and core compound **51** was obtained in 86% yield, which further reacted with 5-substituted tryptamines **42–47** to give final type 1 products with a yield of more than 50%. The synthesis of compounds **1–6** was conducted under microwave irradiation, similar to our previous reports [15].

Type 2 compounds were synthesized according to Figure 3. First, amines **61**–**68** were obtained in the reaction of 2-chloroethylamine hydrochloride **52** and appropriately substituted aniline **53**–**60**. Subsequently, the resulting compounds reacted with readily available [20] core compound **69** in the presence of K_2_CO_3_ and microwave irradiation (*p* = 50 W) for 2.5 min [15]. Final type 2 compounds were isolated with yields of 43–75%. In spite of the complete conversion of substrate **69** (according to TLC), compounds **8**, **12,** and **15** were not obtained. However, it was found that the ethyl bridge was probably eliminated during the reaction (see the mass spectra, Appendix A) to give compounds **20**–**22** (Figure 1, type 4) in which aniline derivatives were attached directly to the triazine system (Figure 4). The resulting products were easily isolated during work-up as white precipitates, which did not require purification. Their structures were confirmed using spectroscopy: ^1^H NMR, ^13^C NMR, and MS. To confirm our hypothesis, we decided to synthesize the selected compound **21** starting from intermediate **69** and *para*-fluoroaniline (Figure 5). The reaction was conducted similarly to the previous one in the presence of potassium carbonate (3 eq.) and sodium carbonate (3 eq.) to give title compound **21**. According to HPLC-MS analysis, the content of the desired product in the reaction mixture was 23% for potassium carbonate and 73% for sodium carbonate (see Appendix A). Desired compounds **8**, **12**, and **15** were obtained with a yield of more or equal to 60% using the same synthesis method while only changing the base to sodium carbonate.

The synthesis of type 3 compounds (Figure 6) started with protecting 2-chloroethylamine hydrochloride **52** with a Boc group, followed by coupling of the resulting product **70** with phthalimide **71** or benzimidazole **72** and its derivatives **73** and **74**. The reactions were performed under microwave irradiation (P = 60 W) for 60 s in the presence of sodium hydroxide and TBAB (tetrabutylammonium bromide). Isolated products **75**–**78** reacted with **69** under microwave irradiation (P = 50 W) in the presence of K_2_CO_3_ and TBAB to give final compounds **15**–**18** with yields of 25–56%.

### 2.2. Radioligand Binding and SAR

The affinity of the studied compounds to 5-HT_1A_, 5-HT_2A_, 5-HT_6_, 5-HT_7_, and D_2_ receptors was evaluated by a radioligand assay, as reported previously (Table 1) [17]. Each compound was tested in triplicate at seven concentrations (from 0.1 nM to 100 µM). Inhibition constants (*K*_i_) were calculated from the Cheng–Prusoff equation [23].

The substituent at indole position 5 in the A ring (type 1 compound) had no significant effect on increased binding to the 5-HT_7_ receptor. The ligand with a weakly deactivating fluorine substituent **2** was found to be the most active among the tested type 1 compounds and showed an affinity of *K*_i_ = 8 nM toward 5-HT_7_R. In addition, in terms of the effect of the halogen atom, ligand **3** with bromine (5-HT_7_R *K*_i_ = 126 nM) had moderate activity, followed by ligand **4** with chlorine (5-HT_7_R, *K*_i_ = 481 nM). Indole substitution at position 5 with CN, a strongly deactivating substituent (compound **1**), resulted in activity toward the 5-HT_7_ receptor being lost. As for two activating substituents: Me (weakly activating) and OMe (strongly activating), moderate activity toward the 5-HT_7_ receptor was shown for ligand **6** (*K*_i_ = 100 nM), while ligand **5** had a low activity with *K*_i_ = 629 nM. As for type 1 compounds, a halogen substituent at the B ring resulted in active (**9**, **11**, **12**) or moderately active ligands (**7**, **8**, **10**) with respect to 5-HT_7_R, while a slightly larger substituent (OMe) resulted in the loss of activity. When analyzing the effect of the substituted position, ligands in the *para* position had the highest activity, and the *ortho* position was the least active. It is concluded based on analysis of the data that the fluorine substituent (*para* > *meta* > *ortho*) is the strongest, followed by chlorine (*para* > *meta* > *ortho*) and finally methoxy (*para* > *meta* > *ortho*). For example, *para*-F (**18**) and *para*-Cl (**9**), with *K*_i_ values of 18 nM and 19 nM, respectively, were found to be the most active ligands. Isomeric *meta*-F (**11**) and *meta*-Cl (**8**) compounds had slightly lower activity compared to the previous ones, with *K*_i_ values of 24 nM and 131 nM, respectively. Even though the compounds with the OMe substituent were inactive, a tendency for a relatively stronger *para* position compared to the weaker *meta* and, finally, the weakest ortho position was also seen in this group. As for type 3 ligands, incorporation of larger substituents R_3_ resulted in most cases in the loss of activity (*K*_i_ > 1000 nM) toward 5-HT_7_R (ligands **16**, **18**, and **19**). Only the ligand with an unsubstituted benzimidazole ring (**17**) had a moderate activity with *K*_i_ = 227 nM. The type 4 compounds are also found to be inactive (**20**, **21**) or weakly active (**22**), but they provide further evidence of the significant effect of the linker between the triazine core and the aromatic system [20]. When analyzing affinity to the other receptors tested (5-HT_1A_, 5-HT_2A_, 5-HT_6,_ and D_2_), most of the designed compounds have no activity (*K*_i_ > 1000 nM) or low activity (500 nM < *K*_i_ < 1000 nM).

### 2.3. Atlas Activity Analysis: 3D-SAR

To better understand SAR, we determined specific activity maps as a function of shape, hydrophobicity, and electrostatics using Activity Atlas (Flare, Cresset) [24]. According to the ligand shape, the incorporation of large substituents at indole position 5 (A ring) results in reduced activity toward 5-HT_7_R. A similar effect is found for substituents at the alkylaromatic ring (B ring). It can be concluded based on Figure 3B,D that large substituents (ligands **13**–**19**, *K*_i_ > 226 nM) cannot be accommodated in the receptor binding pocket, resulting in the loss of activity as shown by an increased number of steric clashes found between the ligand and the binding pocket. Ligands **7**–**12** with small substituents such as chlorine or fluorine show a much better fit to the binding pocket (lower number of steric clashes) as shown by their high or moderate activity to the 5-HT_7_ receptor (Figure 3A,C). It is noted that no steric clashes were found between the substituent and the receptor binding pocket for the most active type 2 ligand **12** (*K*_i_ = 18 nM). This is due to the fact that ligands at the para positions are most preferred with the best fit to the receptor.

According to ligand hydrophobicity, type 2 ligands (ligands **7**–**12**) show the best fit to the favorable hydrophobic region (squared green) due to the presence of a hydrophobic aromatic ring which may contribute to π–π interactions (B ring, Figure 4A). The unfavorable hydrophobic region (squared magenta) is not occupied by hydrophobic substituents. Type 3 ligands (ligands **16**–**19**) are a different case. The favorable hydrophobic region is occupied by hydrophilic parts of the substituents (imide and imidazole systems), while the unfavorable hydrophobic region is occupied by the hydrophobic aromatic ring (Figure 4B). The effect of the hydrophobic/hydrophilic properties of substituents at indole position 5 (A ring) for type 1 ligands was difficult to determine based on our studies.

### 2.4. Molecular Modelling

The highly active compounds (**2**
*K*_i_ = 8 nM and **12**
*K*_i_ = 18 nM) were selected for studying their bioconformation and binding modes through molecular modeling using Induced Fit Docking (Schrodinger, Maestro [25]) followed by ligand–receptor complex optimization using a QM-MM mixed quantum mechanical method (Schrodinger, Maestro [25]) [15,20]. Both ligands occupy a binding pocket typical of 5-HT_7_ receptor ligands [26]. They are oriented toward the inside of the receptor (between TMh3 and TMh5, TMh–*transmembrane helix*) with the tryptamine part, while the alkylaromatic part faces the external receptor surface (between TMh2 and TMh6). Both ligands have more linear bioconformation (Figure 5) with a clear bend of the alkylaromatic moiety and almost completely overlapping hydrogen bonds with the following amino acids: Glu366 (E7.34), Asp162 (D3.32), Ile233, Ser243 (S5.42) (for ligand **2**), and π–π stacking hydrophobic interactions with the following amino acids: Trp340 (W6.48) and Phe343 (F6.51). The binding mode for ligands **2** and **12** corresponds to the results reported previously [15,17,20,26].

### 2.5. Metabolic Stability

Highly active compounds **2** and **12** were submitted for metabolic stability evaluation tests using mouse liver microsomes (MLMs). UHPLC-MS analysis of the samples after compounds **2** and **12** were incubated for 2 h in the presence of MLMs showed that 20% and 29% of the original compound remained, respectively. Table 2 shows potential metabolic pathways for the tested compounds as well as Figure 6 shows possible main metabolites predicted by Metasite 6.0.1. It was found, based on the results, that the presence of fluorine at indole position C-5 (ligand **2**) did not have a significant effect on increased metabolic stability compared to the unsubstituted compound [20]. It seems, however, that substituents at the B ring of the ligands have a more important role. Compound **12,** containing a fluorine atom, was found to be more stable than compound **2** and the reference verapamil. The data correspond to our previous results [15], in which substituents at the B ring increased metabolic stability. Potential sites in the molecules sensitive to enzyme activity and leading to degradation were proposed using an in silico approach, also with MetaSite 6.0.1 (Figure 7).

### 2.6. CYP3A4 Interaction

Potential drug–drug interactions (DDIs) are an important aspect that should be considered when designing new compounds. Isoenzyme CYP3A4 is one of the varieties of enzymes responsible for xenobiotic metabolism [28]. We evaluated active compounds **2** and **12** in terms of affinity toward this isoenzyme (Figure 8). Both compounds at concentrations identical to that of ketoconazole, the reference compound (1 µM) or lower were shown to have no or slight inhibition activity toward CYP3A4. Compound **2** at 10 µM showed moderate (%CYP3A4 activity = 50) and compound **12** showed high (%CYP3A4 activity = 20) inhibition effect, respectively. A very high inhibition effect was observed for both compounds at 25 µM (%CYP3A4 activity < 15).

### 2.7. Hepatotoxicity

The tested compounds **2** and **12** were evaluated in terms of cytotoxicity against the HepG2 cell line to assess their hepatotoxic potential (Figure 9). It was an interesting finding that both compounds had proliferative activity in lower concentrations (<10 µM). A cytotoxic effect appeared only at 50 µM (**2**, % cell viability = 35; **12**, % cell viability = 20); at 100 µM, the cells were practically no viable.

### 2.8. In Vivo Cardiotoxicity

The compounds were evaluated for their ecotoxicity on the Danio rerio experimental model. OECD 236 test [29] with modifications was applied. Both compounds were found cardiotoxic within the non-toxic range, based on the heart rate measurement. For the compound **2**, 5 µg/mL was found cardiotoxic (Figure 10) and for **12**, 7.5 µg/mL (Figure 11). The results were confirmed by malformations observation. Pericardial edema (PE) was observed at 5 µg/mL for **2** (Figure 12) and at 7.5 µg/mL for **12** (Figure 13). Moreover, scoliosis (S) and tail autophagy (TA) were also noted.

## 3. Discussion and Conclusions

In spite of the significant functions of the 5-HT_7_ receptor [30] both in and outside the central nervous system and the recent progress in medicinal chemistry and pharmacology, a drug having selectivity toward 5-HT_7_R is yet to be fully developed [30]. Taking this into consideration, the objective of this paper was to design ligands showing high activity and selectivity toward the 5-HT_7_ receptor without incorporating the arylpiperazine pharmacophore, which may increase affinity to other aminergic GPCRs.

All finally synthesized compounds were obtained by a condensation reaction supported by microwave irradiation for 2.5 min with a yield of more than 50%. An interesting fact turned out to be the synthesis of ligands **8**, **12,** and **15**. So far, working on 1,3,5-triazines with indole motif [15,20], we have successfully used a mild alkaline agent, which was potassium carbonate, obtaining final products with medium or high yield. In the case of the mentioned ligands, the reactions did not proceed as it was expected, leading to the elimination of the ethyl bridge. The usage of a slightly weaker base, which was sodium carbonate, resulted in obtaining the desired products with good yield. It was found when studying type 1 compounds that incorporation of EWG or EDG substituents in most cases did not improve affinity toward the 5-HT_7_ receptor in the majority of synthesized compounds. Compounds **1** and **3**–**6** had lower activity than the unsubstituted compound (Figure 1A). Ligand **2,** whose activity did not change compared to the unsubstituted compound, was an exception (Figure 1A). The fact may be accounted for by the effect of bioisosterism of the fluorine atom (hydrogen bioisostere). When exploring the aromatic region of the B ring, ligands with chlorine, fluorine, and methoxy substituents in the *ortho* position showed the lowest activity; higher activity was shown for the substituents in the *meta* position and the highest for substituents in the *para* position (ligand **12**, *K*_i_ = 18 nM). It is noted that ligands with small substituents (Cl, F) were more active than those with slightly larger substituents (OMe). The incorporation of much larger heterocyclic compounds (type 3) resulted in the loss of activity on the 5-HT_7_ receptor. Type 4 compounds showed once more that the distance between the triazine core and the aromatic system, which should be two or three atoms, was crucial [15,20]. The SARs for the resulting compounds were supported using 3D-QSAR computed methods. The bioconformation and the binding mode for the two best compounds (**2** and **12**) were determined using molecular modeling (docking), and the results were consistent with our previous reports [15,20]. The tested ligands (**2** and **12**) are 5-HT_7_ receptor antagonists, have moderate metabolic activity (higher or similar to verapamil and still higher than unsubstituted ligands which were described in our publication [20]), and moderate or weak potential drug-drug interactions with respect to ketoconazole. As for hepatotoxicity, both compounds at more than 50 µM showed cytotoxicity against the HepG2 cell line. The ecotoxicity tests with the use of Daio rerio as a model organism turned out **2** to be more toxic than **12**. Moreover, cardiotoxicity expressed as heart rate abnormalities was observed at higher doses for **2** than **12,** and it suggests that cardiotoxic potential needs to be reduced in the future.

## 4. Materials and Methods

### 4.1. Chemistry

#### 4.1.1. General

All primary substrates were purchased commercially from Sigma-Aldrich (St. Louis, MO, USA). The solvents used for column chromatography (purchased from Merck, Kenilworth, NJ, USA), thin layer chromatography (TLC), and preparative thin layer chromatography (pTLC) had purity above 99.5%. ^1^H and ^13^C NMR spectra were recorded using Bruker 400 MHz systems with TMS as an internal standard. Melting points were determined with the Böetius apparatus. HPLC–MS analyses were performed on the Shimadzu Nexera XR system equipped with PDA (SPD-M40) and LCMS-2020 detectors. Analyses were performed on Phenomenex XB-C18 1.7 μm (50 × 2.1 mm) (method A) column with gradient of solvents as a mobile phase: Solvent A (0.01% HCOOH in water) and B (0.01% HCOOH in methanol); t = 0 min, 10% of B, t = 4 min, 90% of B, t = 6 min, 90% of B, t = 6.1 min 10% of B, stop time 11 min or Phenomenex C18 1.7 μm (50 × 2.1 mm) (method B) column with gradient of solvents as a mobile phase: solvent A (0.01% HCOOH in water) and solvent B (0.01% HCOOH in MeOH): t = 0 min 5% of B, t = 3 min 90% of B, t = 4 min 90% of B, t = 4.5 min 5% of B stop time 7 min; flow rate 0.4 mL min^−1^; the UV-VIS detection was performed in a range of 240–700 nm, the MS data were collected in ESI + mode in a range of *m*/*z* 100–800 with scan speed 15,000 u/s and event time 0.1 s. Analytical thin-layer chromatography (TLC) was performed using 0.2 mm silica gel precoated aluminum sheets (60 F254, Merck), and UV light at 254 nm was used for visualization. Preparative thin-layer chromatography (pTLC) was performed using 2000 µm silica gel precoated glass backed (F254, Silicycle). A CEM Discover™ Focused Microwave System at 50 W power was used for all microwave-assisted reactions in order to obtain final compounds. Within 2.5 min of reaction with a power of 50 W, the temperature increased up to 120 °C, while the pressure increased up to 9 bar. Characterization of the intermediates and spectra for the final compounds can be found in Appendix A.

#### 4.1.2. General Procedure for the Synthesis of Compounds **29**–**34**

12 mL of DMF was cooled to 0 °C, and phosphoryl chloride (38.5 mmol) was added dropwise. In this temperature solution of commercially available indoles **23**–**28** (35.5 mmol) in 3 mL of DMF was added dropwise, and the resulting mixture was stirred at room temperature for one hour. The reaction became a thick, pale suspension. Sodium hydroxide solution (10%, 40 mL) was added slowly to the reaction mixture (pH = 13–14), followed by precipitation of pale solid **29**–**34**. The solid was filtered, rinsed with distilled water, and dried.

#### 4.1.3. General Procedure for the Synthesis of Compounds **35**–**40**

Intermediate **29**–**34** (23.5 mmol) was placed in a one-necked round bottom flask and dissolved in 60 mL nitromethane, and then, ammonia acetate (44.2 mmol) was added. The mixture was reacted in a microwave reactor at 80 W (100 °C) for 20 min. Reaction progress was monitored via TLC (hexane:EtOAc 1:1 *v*/*v*). After this time, TLC indicated full conversion of starting material, and the mixture was cooled to room temperature with precipitation of yellow solid **35**–**40**. The solid was filtered, rinsed with distilled water, and dried.

#### 4.1.4. Synthesis of 3-(2-Nitroethyl)-1H-Indole-5-Carbonitrile (**41**)

Intermediate **35** (4.1 g, 19.2 mmol) was dissolved in the 330 mL mixture of DMF:MeOH (1:1, *v*/*v*), followed by the addition of sodium borohydride (8 g, 21.1 mmol). The mixture was stirred at room temperature for 5 h. Reaction progress was monitored via TLC (hexane:EtOAc 1:1 *v*/*v*). After reaction completion reaction was diluted with 2M HCl to reach pH = 7. The solvent was reduced and extracted with EtOAc (3 × 150 mL). The crude product was triturated with a mixture of MeOH:chloroform to yield 2.17 g of the titled compound. The mother liquor was concentrated to dryness and purified at column chromatography eluted with hexane:EtOAc (*v*/*v*) 8:2 -> 6:4 to yield 0.8 g of the titled compound. Creamy solid (71% yield); mp.: 131–133 °C (ref. 134–136 °C [31]); method B: ESI+MS calc. for C_11_H_9_N_3_O_2_ *m*/*z* = 215; found *m*/*z* = 214 [M-H]^−^.

#### 4.1.5. General Procedure for the Synthesis of Compounds **42**–**46**

Lithium aluminum hydride (25.9 mmol) was placed in a three-necked round bottom flask, followed by the addition of 20 mL dry THF. The resulting suspension was cooled to 0 °C, and mixture of intermediates **36**– **40** (4.7 mmol) in 20 mL THF was added dropwise. The mixture was stirred at room temperature for 72 h and then quenched with slow addition of mixture H_2_O:MeOH (9:1, *v*/*v*). The suspension was filtered by a Celite bed, and pH-dependent extraction was performed: filtrate was acidified to reach pH 2–3 with 1 M HCl, followed by extraction with EtOAc (3 × 100 mL). The water layer was alkalized to reach pH 10 with 1 M NaOH and then extracted with EtOAc (3 × 100 mL). Organic layers were combined, dried over MgSO_4_, and concentrated to yield brown, sticky oil **42**–**46**.

#### 4.1.6. Synthesis of 3-(2-Aminoethyl)-1H-Indole-5-Carbonitrile (**47**)

A solution of **41** (1.5 g, 6.9 mmol) in 205 mL MeOH was added to the solution of zinc (10.4 g, 0.16 mol) in 205 mL 2M HCl and refluxed for 1.5 h. After this time mixture was cooled to room temperature and filtered. The filtrate was alkalized to pH 12, and MeOH was removed under reduced pressure. The resulting mixture was extracted with EtOAc (3 × 200 mL), and organic layers were combined, dried over MgSO_4_ then evaporated to dryness. It obtained 0.87 g (67% yield) of titled compound **47,** which was used in the next step without any further purification.

#### 4.1.7. Synthesis of 4,6-Dichloro-N-Phenethyl-1,3,5-Triazin-2-Amine (**50**)

To a solution of cynuric chloride **48** (8.35 g, 45.2 mmol) in 100 mL THF cooled to 0 °C, a solution of phenylethylamine **49** (5 g, 41.2 mmol) in 5 mL THF was added dropwise. The reaction was carried out at 0–3 °C for 2 h. The resulting precipitate was filtered, and the filtrate was diluted with 0.1 M HCl and extracted with chloroform (3 × 100 mL). Organic layers were combined, dried over MgSO_4_, and concentrated to yield a brown solid. The solid was triturated with acetone and then filtered. The black filtrate was purified using column chromatography eluted with hexane:EtOAc (*v*/*v*) 9:1-> 6:4 to yield 3.74 g of the titled compound. Creamy solid (34% yield); method B: ESI-MS calc. for C_11_H_10_Cl_2_N_4_ *m*/*z* = 268.0; found *m/z* = 269.1 [M+H]^+^.

#### 4.1.8. Synthesis of 6-Chloro-N^2^-Phenethyl-1,3,5-Triazine-2,4-Diamine (**51**)

Briefly, **50** (3.5 g, 13.0 mmol) was dissolved in 50 mL acetone, followed by the addition of 5.7 mL 25% ammonia solution. The reaction was carried out at room temperature for 5 h. The resulting precipitate was filtered, and the filtrate evaporated to dryness, yielding 2.8 g of titled compound **51**. Creamy solid (86% yield); ESI-MS calc. for C_11_H_12_ClN_5_ *m/z* = 249; method B: found *m*/*z* = 250 [M+H]^+^.

#### 4.1.9. General Procedure for the Synthesis of Final Compounds **1**–**6** (Microwave-Assisted)

Briefly, **51** (0.25 g, 1.0 mmol), potassium carbonate (0.41 g, 3.0 mmol), and TBAB (0.032 g, 0.1 mmol) were ground in a mortar and transferred to a sealed tube which was previously charged with appropriate amine **42**–**47** (2.5 mmol). Subsequently, 5 wt % DMF was added. The mixture was reacted in a microwave reactor at 50 W for 2.5 min. Reaction progress was monitored via TLC (chloroform: MeOH 9:1 *v*/*v*). The mixture was cooled down and extracted with chloroform (3 × 20 mL). Organic layers were combined, dried over MgSO_4_, and concentrated. The crude product was purified via column chromatography with elution using chloroform then chloroform:MeOH (*v*/*v*) 99:1-> 97:3. The white or beige sticky oil was then dissolved in acetone and pH was adjusted to 2–3 with 4 M HCl in 1,4-dioxane. The resulting mixture was crushed by the addition of cold diethyl ether, then the white or beige powder was filtered and rinsed with cold diethyl ether and then dried to yield final product **1**–**6**.

#### 4.1.10. 3-(2-((4-Amino-6-(Phenethylamino)-1,3,5-Triazin-2-yl)Amino)Ethyl)-1H-Indole-5-Carbonitrile Hydrochloride (**1**)

Beige solid (54% yield), mp: 95–98 °C; ^1^H NMR (600 MHz, MeOD) δ 8.02 (s, 1H), 7.51 (d, J = 8.4 Hz, 1H), 7.40 (t, J = 9.9 Hz, 1H), 7.34–7.15 (m, 6H), 3.76–3.70 (m, 2H), 3.64 (t, J = 7.0 Hz, 1H), 3.59 (t, J = 7.1 Hz, 1H), 3.09 (t, J = 7.0 Hz, 2H), 2.91 (d, J = 7.3 Hz, 1H), 2.86 (t, J = 7.1 Hz, 1H); ^13^C NMR (151 MHz, MeOD) δ 156.0, 138.6, 138.5, 128.4, 128.1, 128.1, 127.3, 126.0, 125.1, 125.1, 123.8, 123.7, 120.5, 112.9, 112.1, 100.9, 42.0, 41.3, 35.0, 24.5; HPLC-MS analysis t–5.84 min (94% purity, method A), calc. for C_22_H_22_N_8_ *m/z* = 398.2, found *m*/*z* = 399.2 [M+H]^+^.

#### 4.1.11. N^2^-(2-(5-Fluoro-1H-Indol-3-yl)ethyl)-N^4^-Phenethyl-1,3,5-Triazine-2,4,6-Triamine Hydrochloride (**2**)

Beige solid (67% yield), mp: 110–114 °C; ^1^H NMR (600 MHz, MeOD) δ 7.33–7.29 (m, 2H), 7.27–7.14 (m, 6H), 6.87 (t, J = 7.9 Hz, 1H), 3.76–3.67 (m, 2H), 3.63 (t, J = 6.2 Hz, 1H), 3.60 (t, J = 7.2 Hz, 1H), 3.03 (t, J = 6.5 Hz, 2H), 2.90 (t, J = 6.8 Hz, 1H), 2.86 (t, J = 7.1 Hz, 1H); ^13^C NMR (151 MHz, MeOD) δ 158.2, 156.7, 156.1, 133.3, 132.2, 131.0, 128.5, 128.1, 126.0, 124.3, 111.7, 109.1, 108.9, 102.5, 102.3, 42.0, 41.3, 35.0, 24.8; HPLC-MS analysis t–6.13 min (99% purity, method A), calc. for C_21_H_22_FN_7_ *m/z* = 391.2, found *m/z* = 392.2 [M+H]^+^.

#### 4.1.12. N^2^-(2-(5-Bromo-1H-Indol-3-yl)ethyl)-N^4^-Phenethyl-1,3,5-Triazine-2,4,6-Triamine Hydrochloride (**3**)

Beige solid (63% yield), mp: 82–85 °C; ^1^H NMR (600 MHz, MeOD) δ 7.70 (d, J = 13.1 Hz, 1H), 7.33–7.17 (m, 8H), 3.71 (t, J = 7.2 Hz, 2H), 3.64 (t, J = 7.2 Hz, 1H), 3.60 (t, J = 7.1 Hz, 1H), 3.02 (dd, J = 12.8, 5.8 Hz, 2H), 2.91 (t, J = 7.2 Hz, 1H), 2.87 (t, J = 7.0 Hz, 1H); ^13^C NMR (151 MHz, MeOD) δ 156.1, 138.7, 135.4, 129.2, 128.4, 128.1, 128.1, 126.1, 126.0, 123.9, 123.6, 120.4, 112.6, 111.5, 111.4, 66.98 47.2, 42.0, 41.4, 35.0, 24.7; HPLC-MS analysis t–6.41 min (98% purity, method A), calc. for C_21_H_22_^79^BrN_7_ *m/z* = 451.1, found *m/z* = 452.2 [M+H]^+^, calc. for C_21_H_22_^81^BrN_7_ *m/z* = 453.1, found *m/z* = 454.2 [M+H]^+^.

#### 4.1.13. N^2^-(2-(5-Chloro-1H-Indol-3-yl)ethyl)-N^4^-Phenethyl-1,3,5-Triazine-2,4,6-Triamine Hydrochloride (**4**)

White solid (58% yield), mp: 106–108 °C; ^1^H NMR (600 MHz, MeOD) δ 7.55 (d, J = 12.2 Hz, 1H), 7.32 (dd, J = 17.0, 8.0 Hz, 2H), 7.28–7.15 (m, 5H), 7.07 (dd, J = 8.5, 1.7 Hz, 1H), 3.74–3.67 (m, 2H), 3.64 (t, J = 7.2 Hz, 1H), 3.60 (t, J = 7.1 Hz, 1H), 3.04 (t, J = 7.0 Hz, 2H), 2.91 (t, J = 7.3 Hz, 1H), 2.87 (t, J = 7.1 Hz, 1H); ^13^C NMR (151 MHz, MeOD) δ 156.0, 138.6, 135.1, 128.5, 128.5, 128.4, 128.1, 128.1, 126.0, 124.0, 121.1, 117.3, 117.2, 112.1, 111.4, 42.0, 41.4, 35.0, 24.7; HPLC-MS analysis t–6.11 min (94% purity, method A), calc. for C_21_H_22_^35^ClN_7_ *m*/*z* = 407.2, found *m*/*z* = 408.4 [M+H]^+^, calc. for C_21_H_22_^37^ClN_7_ *m*/*z* = 409.2, found *m*/*z* = 410.4 [M+H]^+^.

#### 4.1.14. N^2^-(2-(5-Methoxy-1H-Indol-3-yl)ethyl)-N^4^-Phenethyl-1,3,5-Triazine-2,4,6-Triamine Hydrochloride (**5**)

White solid (52% yield), mp: 111–115 °C; ^1^H NMR (600 MHz, MeOD) δ 7.31 (t, J = 7.4 Hz, 2H), 7.27–7.21 (m, 4H), 7.18 (d, J = 7.5 Hz, 1H), 7.07 (t, J = 20.0 Hz, 1H), 6.78 (t, J = 8.3 Hz, 1H), 3.83 (d, J = 4.2 Hz, 1H), 3.76 (s, 3H), 3.70 (t, J = 6.7 Hz, 1H), 3.66–3.61 (m, 1H), 3.57 (t, J = 7.3 Hz, 1H), 3.05 (dd, J = 14.5, 7.3 Hz, 2H), 2.90 (t, J = 7.2 Hz, 1H), 2.85 (t, J = 7.3 Hz, 1H); ^13^C NMR (151 MHz, MeOD) δ 156.0, 153.5, 138.6, 132.0, 128.4, 128.1, 128.1, 127.6, 126.1, 126.0, 123.0, 111.5, 111.3, 111.2, 111.1, 54.9, 42.0, 41.5, 35.0, 24.8; HPLC-MS analysis t–5.91 min (98% purity, method A), calc. for C_22_H_25_N_7_O *m*/*z* = 403.2, found *m*/*z* = 404.2 [M+H]^+^.

#### 4.1.15. N^2^-(2-(5-Methyl-1H-Indol-3-yl)ethyl)-N^4^-Phenethyl-1,3,5-Triazine-2,4,6-Triamine Hydrochloride (**6**)

White solid (57% yield), mp: 118–121 °C; ^1^H NMR (600 MHz, MeOD) δ 7.31 (dd, J = 15.8, 8.7 Hz, 2H), 7.28–7.21 (m, 4H), 7.18 (t, J = 6.2 Hz, 1H), 7.06 (d, J = 21.7 Hz, 1H), 6.94 (dd, J = 12.1, 8.5 Hz, 1H), 3.75 (t, J = 7.1 Hz, 1H), 3.70 (bs, 1H), 3.63 (bs, 1H), 3.56 (t, J = 7.3 Hz, 1H), 3.05 (t, J = 6.7 Hz, 2H), 2.90 (t, J = 7.2 Hz, 1H), 2.85 (t, J = 7.3 Hz, 1H), 2.38 (s, 3H)–hydrogen bonded rotamer H_3_C-C_Ar_; ^13^C NMR (151 MHz, MeOD) δ 156.0, 138.6, 135.1, 128.5, 128.4, 128.1, 128.1, 127.6, 127.3, 126.1, 126.0, 122.6, 122.3, 117.4, 110.6, 42.0, 41.5, 35.1, 24.9, 20.3 (H_3_C-C_Ar_). HPLC-MS analysis t–6.00 min (98% purity, method A), calc. for C_22_H_25_N_7_ *m/z* = 387.2, found *m/z* = 388.4 [M+H]^+^.

#### 4.1.16. General Procedure for the Synthesis of Compounds **61**–**68**

In a round bottom flask, 2-chloroethanamine hydrochloride **52** (1.0 g, 8.6 mmol) was suspended in 8 mL of toluene. To the resulting mixture, appropriate aniline **53**–**60** (51.7 mmol) was added, and the mixture was refluxed for 20 h. After this period, toluene was evaporated, and residues were triturated with dichloromethane to yield solid as titled compounds **61**–**68** (optionally solid may be washed with diethyl ether).

#### 4.1.17. General Procedure for the Synthesis of Final Compounds **7**, **9–11**, and **14** (microwave-assisted)

Briefly, **69** [20] (0.25 g, 0.8 mmol), amines **61**, **63**–**65, 67** (2.0 mmol) potassium carbonate (0.36 g, 2.5 mmol) and TBAB (0.032 g, 0.1 mmol) were ground in a mortar and transferred to a sealed tube. Subsequently, 5 wt % DMF was added. The mixture was reacted in a microwave reactor at 50 W for 2.5 min. Reaction progress was monitored via TLC (chloroform: MeOH 9:1 *v*/*v*). The mixture was cooled down and extracted with chloroform (3 × 20 mL). Organic layers were combined, dried over MgSO_4_, and concentrated. The crude product was purified via column chromatography with elution using chloroform then chloroform:MeOH (*v*/*v*) 99:1-> 97:3. Colorless sticky oil was then dissolved in acetone, and pH was adjusted to 2–3 with 4 M HCl in 1,4-dioxane. The resulting mixture was crushed by the addition of cold diethyl ether, and then the white or beige powder was filtered and rinsed with cold diethyl ether and then dried to yield final products **7**, **9**–**11,** and **14**.

#### 4.1.18. N^2^-(2-(1H-Indol-3-yl)ethyl)-N^4^-(2-((2-Chlorophenyl)amino)ethyl)-1,3,5-Triazine-2,4,6-Triamine Hydrochloride (**7**)

White solid (72% yield), mp: 105–108 °C; ^1^H NMR (600 MHz, MeOD) δ 7.56 (dd, J = 19.1, 7.7 Hz, 1H), 7.35 (d, J = 8.1 Hz, 1H), 7.33–6.99 (m, 5H), 6.99–6.87 (m, 1H), 6.82–6.66 (m, 1H), 3.72 (dd, J = 15.3, 7.2 Hz, 2H), 3.66 (dd, J = 13.3, 6.3 Hz, 1H), 3.54 (t, J = 6.0 Hz, 1H), 3.49 (t, J = 5.9 Hz, 1H), 3.41 (t, J = 6.1 Hz, 1H), 3.07 (t, J = 6.4 Hz, 1H), 3.03 (t, J = 7.0 Hz, 1H); ^13^C NMR (151 MHz, MeOD) δ 156.0, 142.0, 136.8, 129.1, 127.7, 127.3, 122.3, 121.0, 119.8, 118.9, 118.3, 118.2, 117.7, 112.8, 111.3, 110.9, 43.3, 41.4, 38.8, 24.8; HPLC-MS analysis t–5.92 min (98% purity, method A), calc. for C_21_H_23_^35^ClN_8_ *m*/*z* = 422.2, found *m*/*z* = 423.2 [M+H]^+^, calc. for C_21_H_23_^37^ClN_8_ *m*/*z* = 424.2, found *m*/*z* = 425.2 [M+H]^+^.

#### 4.1.19. N^2^-(2-(1H-Indol-3-yl)ethyl)-N^4^-(2-((4-Chlorophenyl)amino)ethyl)-1,3,5-Triazine-2,4,6-Triamine Hydrochloride (**9**)

White solid (70% yield), mp: 97–100 °C; ^1^H NMR (600 MHz, MeOD) δ 7.58 (d, J = 7.8 Hz, 1H), 7.54–7.46 (m, 2H), 7.43 (d, J = 8.7 Hz, 1H), 7.36 (d, J = 8.0 Hz, 2H), 7.12 (dd, J = 18.5, 13.7 Hz, 2H), 7.03 (dt, J = 14.8, 7.5 Hz, 1H), 3.79–3.73 (m, 2H), 3.71 (t, J = 6.9 Hz, 1H), 3.58 (d, J = 5.2 Hz, 1H), 3.56 (d, J = 5.2 Hz, 1H), 3.16 (d, J = 18.8 Hz, 1H), 3.09 (t, J = 7.0 Hz, 1H), 3.04 (t, J = 6.9 Hz, 1H); ^13^C NMR (151 MHz, MeOD) δ 156.1, 136.8, 129.9, 127.3, 123.0, 122.8, 122.4, 122.3, 121.0, 118.3, 117.7, 117.7, 111.2, 111.0, 110.9, 41.3, 41.2, 36.8, 24.8; HPLC-MS analysis t–5.94 min (92% purity, method A), calc. for C_21_H_23_^35^ClN_8_ *m*/*z* = 422.2, found *m*/*z* = 423.4 [M+H]^+^, calc. for C_21_H_23_^37^ClN_8_ *m*/*z* = 424.2, found *m*/*z* = 425.4 [M+H]^+^.

#### 4.1.20. N^2^-(2-(1H-Indol-3-yl)ethyl)-N^4^-(2-((2-Fluorophenyl)amino)ethyl)-1,3,5-Triazine-2,4,6-Triamine Hydrochloride (**10**)

White solid (70% yield), mp: 94–97 °C; ^1^H NMR (600 MHz, MeOD) δ 7.60–7.52 (m, 1H), 7.36 (d, J = 8.1 Hz, 1H), 7.34–6.95 (m, 7H), 3.74 (t, J = 6.8 Hz, 3H), 3.61–3.54 (m, 2H), 3.48 (t, J = 6.1 Hz, 1H), 3.08 (t, J = 6.8 Hz, 1H), 3.04 (t, J = 7.0 Hz, 1H); ^13^C NMR (151 MHz, MeOD) δ 158.5, 156.1, 136.8, 127.3, 127.2, 125.0, 124.9, 122.4, 122.3, 121.0, 118.2, 117.7, 115.4, 115.3, 111.3, 110.9, 45.4, 41.3, 38.0, 24.8; HPLC-MS analysis t–5.79 min (99% purity, method A), calc. for C_21_H_23_FN_8_ *m*/*z* = 406.2, found *m*/*z* = 407.4 [M+H]^+^.

#### 4.1.21. N^2^-(2-(1H-Indol-3-yl)ethyl)-N^4^-(2-((3-Fluorophenyl)amino)ethyl)-1,3,5-Triazine-2,4,6-Triamine Hydrochloride (**11**)

White solid (52% yield), mp: 74–79 °C; ^1^H NMR (600 MHz, MeOD) δ 7.58 (d, J = 7.8 Hz, 1H), 7.35 (d, J = 8.1 Hz, 1H), 7.10 (t, J = 7.5 Hz, 2H), 7.02 (bs, 2H), 6.46–6.23 (m, 3H), 3.67 (bs, 2H), 3.53 (bs, 2H), 3.28 (t, J = 6.3 Hz, 2H), 3.04 (t, J = 7.1 Hz, 2H); ^13^C NMR (151 MHz, MeOD) δ 165.0, 163.4, 136.8, 129.8, 127.3, 122.2, 120.9, 118.2, 117.9, 111.7, 110.8, 108.2, 102.3, 102.1, 98.5, 98.3, 42.8, 41.1, 39.4, 25.0; HPLC-MS analysis t–5.75 min (94% purity, method A), calc. for C_21_H_23_FN_8_ *m*/*z* = 406.2, found *m*/*z* = 407.3 [M+H]^+^.

#### 4.1.22. N^2^-(2-(1H-Indol-3-yl)ethyl)-N^4^-(2-((3-Methoxyphenyl)amino)ethyl)-1,3,5-Triazine-2,4,6-Triamine Hydrochloride (**14**)

White solid (77% yield), mp: 140–143 °C; ^1^H NMR (600 MHz, MeOD) δ 7.58 (d, J = 7.8 Hz, 1H), 7.51 (d, J = 7.9 Hz, 1H), 7.44 (d, J = 17.4 Hz, 1H), 7.36 (d, J = 7.6 Hz, 2H), 7.17–7.07 (m, 3H), 7.06–6.98 (m, 1H), 3.85 (s, 3H), 3.80 (s, 3H), 3.59 (dd, J = 5.3, 4.0 Hz, 2H), 3.08 (t, J = 6.9 Hz, 2H), 3.03 (t, J = 6.9 Hz, 1H); ^13^C NMR (151 MHz, MeOD) δ 160.9, 156.1, 136.8, 130.7, 130.7, 127.3, 127.2, 122.5, 122.4, 121.0, 118.3, 117.8, 117.7, 111.2, 111.0, 110.9, 54.8, 41.3, 41.2, 36.6, 24.8; HPLC-MS analysis t–5.58 min (99% purity, method A), calc. for C_22_H_26_N_8_O *m*/*z* = 418.2, found *m*/*z* = 419.4 [M+H]^+^.

#### 4.1.23. General Procedure for the Synthesis of Final Compounds **8**, **12**, and **15** (Microwave-Assisted)

Briefly, **69** [20] (0.25 g, 0.8 mmol), amines **62**, **66**, and **68** (2.0 mmol), sodium carbonate (0.26 g, 2.5 mmol) and TBAB (0.032 g, 0.1 mmol) were ground in a mortar and transferred to a sealed tube. Subsequently, 5 wt % DMF was added. The mixture was reacted in a microwave reactor at 50 W for 2.5 min. Reaction progress was monitored via TLC (chloroform: MeOH 9:1 *v*/*v*). The mixture was cooled down and extracted with chloroform (3 × 20 mL). Organic layers were combined, dried over MgSO_4_, and concentrated. The crude product was purified via column chromatography with elution using chloroform then chloroform:MeOH (*v*/*v*) 99:1-> 97:3. Colorless sticky oil was then dissolved in acetone, and pH was adjusted to 2–3 with 4 M HCl in 1,4-dioxane. The resulting mixture was crushed by the addition of cold diethyl ether; then, the white or beige powder was filtered and rinsed with cold diethyl ether then dried to yield final products **8**, **12**, and **15**.

#### 4.1.24. N^2^-(2-(1H-Indol-3-yl)ethyl)-N^4^-(2-((3-Chlorophenyl)amino)ethyl)-1,3,5-Triazine-2,4,6-Triamine Hydrochloride (**8**)

White solid (70% yield), mp: 77–82 °C; ^1^H NMR (600 MHz, MeOD) δ 7.58 (d, J = 7.8 Hz, 1H), 7.35 (d, J = 8.1 Hz, 1H), 7.10 (t, J = 7.8 Hz, 2H), 7.05–6.92 (m, 2H), 6.73–6.51 (m, 3H), 3.68 (s, 2H), 3.53 (s, 2H), 3.28 (t, J = 5.7 Hz, 2H), 3.04 (t, J = 7.1 Hz, 2H); ^13^C NMR (151 MHz, MeOD) δ 150.0, 136.8, 134.4, 129.8, 127.3, 122.2, 122.2, 121.0, 120.9, 118.2, 117.9, 115.8, 111.7, 111.5, 110.8, 110.6, 42.5, 41.1, 39.4, 25.0; HPLC-MS analysis t–5.60 min (100% purity, method A), calc. for C_21_H_23_^35^ClN_8_ *m/z* = 422.2, found *m/z* = 423.2 [M+H]^+^, calc. for C_21_H_23_^37^ClN_8_ *m*/*z* = 424.2, found *m*/*z* = 425.2 [M+H]^+^.

#### 4.1.25. N^2^-(2-(1H-Indol-3-yl)ethyl)-N^4^-(2-((4-Fluorophenyl)amino)ethyl)-1,3,5-Triazine-2,4,6-Triamine Hydrochloride (**12**)

White solid (81% yield), mp: 140–143 °C; ^1^H NMR (600 MHz, MeOD) δ 7.60 (dd, J = 17.7, 5.8 Hz, 2H), 7.52 (dd, J = 20.8, 8.3 Hz, 1H), 7.38–7.28 (m, 2H), 7.22 (t, J = 8.5 Hz, 1H), 7.12 (dd, J = 20.8, 13.1 Hz, 2H), 7.03 (dt, J = 15.0, 7.5 Hz, 1H), 3.79 (d, J = 5.3 Hz, 1H), 3.77–3.72 (m, 2H), 3.69 (d, J = 3.3 Hz, 1H), 3.59 (dd, J = 10.3, 4.5 Hz, 2H), 3.09 (t, J = 7.0 Hz, 1H), 3.05 (t, J = 6.9 Hz, 1H); ^13^C NMR (151 MHz, MeOD) δ 156.2, 136.8, 127.3, 124.3, 122.4, 122.3, 121.0, 118.3, 117.7, 117.7, 116.9, 116.8, 111.2, 111.0, 110.9, 41.3, 41.2, 36.5, 24.8; HPLC-MS analysis t–5.61 min (100% purity, method A), calc. for C_21_H_23_FN_8_ *m*/*z* = 406.2, found *m*/*z* = 407.3 [M+H]^+^.

#### 4.1.26. N^2^-(2-(1H-Indol-3-yl)ethyl)-N^4^-(2-((4-Methoxyphenyl)amino)ethyl)-1,3,5-Triazine-2,4,6-Triamine Hydrochloride (**15**)

White solid (60% yield), mp: 150–153 °C; ^1^H NMR (600 MHz, MeOD) δ 7.59 (d, J = 7.6 Hz, 1H), 7.54–7.44 (m, 2H), 7.40–7.35 (m, 2H), 7.11 (t, J = 6.0 Hz, 2H), 7.03 (dd, J = 19.2, 7.7 Hz, 1H), 6.97 (d, J = 8.9 Hz, 1H), 3.83 (d, J = 13.4 Hz, 2H), 3.75 (s, 3H), 3.71 (t, J = 7.0 Hz, 1H), 3.65 (t, J = 5.6 Hz, 1H), 3.60 (t, J = 5.7 Hz, 1H), 3.56 (t, J = 5.7 Hz, 1H), 3.09 (t, J = 6.6 Hz, 1H), 3.04 (t, J = 6.9 Hz, 1H); ^13^C NMR (151 MHz, MeOD) δ 160.4, 156.2, 136.8, 127.4, 127.3, 127.0, 123.6, 122.4, 121.0, 118.3, 117.7, 115.1, 111.2, 111.0, 110.9, 54.8, 41.35, 36.7, 36.3, 24.8; HPLC-MS analysis t–5.20 min (100% purity, method A), calc. for C_22_H_26_N_8_O *m*/*z* = 418.2, found *m*/*z* = 419.2 [M+H]^+^

#### 4.1.27. Synthesis of Tert-Butyl-(2-Chloroethyl)Carbamate (**70**)

To a suspension of 2-chloroethanamine hydrochloride **52** (5 g, 43.1 mmol) in 60 mL DCM cooled to 0 °C, triethylamine (12 mL, 86.1 mmol) was added, followed by di-tert-butyl decarbonate (11.2 g, 51.4 mmol). The reaction was carried out at room temperature for 12 h. After this period reaction mixture was washed three times with 0.1M HCl and water. Organic layers were combined, dried over MgSO_4_ then evaporated to dryness, yielding colorless sticky oil **70**.

#### 4.1.28. General Procedure for the Synthesis of **75–78** (Microwave-Assisted)

Starting material **71**–**74** (3.7 mmol), tert-butyl-(2-chloroethyl)carbamate **70** (5.6 mmol), sodium hydroxide (5.6 mmol), and TBAB (0.4 mmol) were ground in a mortar and transferred to a sealed tube. Subsequently, 5 wt % DMF was added. The mixture was reacted in a microwave reactor at 65 W for 60 s. Reaction progress was monitored via TLC (chloroform: MeOH 9:1, *v*/*v*). The mixture was extracted with chloroform (3 × 30 mL), organic layers were combined, dried over MgSO_4_ then evaporated to dryness. The crude product was purified via column chromatography, eluting with chloroform then chloroform:MeOH (*v*/*v*) 99:1-> 97:3. Obtained pale yellow solid was then dissolved in 30 mL DCM followed by the addition of 4M HCl in 1,4-dioxane to reach pH = 2. The mixture was stirred at room temperature for 12 h, and then the solid was filtered, washed with DCM and diethyl ether, and dried, yielding titled compounds **75**–**78**.

#### 4.1.29. General Procedure for the Synthesis of Final Compounds **16**–**19** (Microwave-Assisted)

Briefly, **69** [20] (0.25 g, 0.8 mmol), amine hydrochloride **75**–**78** (2.0 mmol), potassium carbonate (0.36 g, 2.5 mmol) and TBAB (0.032 g, 0.1 mmol) were ground in a mortar and transferred to a sealed tube. Subsequently, 5 wt % DMF was added. The mixture was reacted in a microwave reactor at 50 W for 2.5 min. Reaction progress was monitored via TLC (chloroform: MeOH 9:1 *v*/*v*). The mixture was cooled down and extracted with chloroform (3 × 20 mL). Organic layers were combined, dried over MgSO_4_, and concentrated. The crude product was purified via column chromatography with elution using chloroform then chloroform:MeOH (*v*/*v*) 99:1-> 88:12. Beige sticky oil was then dissolved in acetone, and pH was adjusted to 2–3 with 4 M HCl in 1,4-dioxane. The resulting mixture was crushed by the addition of cold diethyl ether, and then the beige powder was filtered and rinsed with cold diethyl ether and then dried to yield final product **16**–**19**.

#### 4.1.30. 2-(2-((4-((2-(1H-Indol-3-yl)ethyl)amino)-6-Amino-1,3,5-Triazin-2-yl)amino)ethyl)isoindoline-1,3-Dione Hydrochloride (**16**)

White solid (59% yield), mp: 136–138 °C; ^1^H NMR (600 MHz, MeOD) δ 7.87–7.66 (m, 4H), 7.56 (d, J = 7.9 Hz, 1H), 7.34 (dd, J = 14.8, 7.9 Hz, 1H), 7.14–6.95 (m, 3H), 3.91 (bs, 1H), 3.84 (t, J = 5.1 Hz, 1H), 3.69–3.50 (m, 4H), 3.07–2.88 (m, 2H); ^13^C NMR (151 MHz, MeOD) δ 168.5, 168.3, 136.7, 133.9, 131.9, 127.3, 122.7, 122.4, 121.0, 120.9, 118.3, 118.1, 117.8, 111.4, 110.8, 41.1, 38.7, 36.9, 24.9; HPLC-MS analysis t–5.41 min (100% purity, method A), calc. for C_23_H_22_N_8_O_2_ *m*/*z* = 442.2, found *m*/*z* = 443.2 [M+H]^+^.

#### 4.1.31. N^2^-(2-(1H-Benzo[d]imidazol-1-yl)ethyl)-N^4^-(2-(1H-Indol-3-yl)ethyl)-1,3,5-Triazine-2,4,6-Triamine Hydrochloride (**17**)

White solid (60% yield), mp: 195–200 °C; 1H NMR (600 MHz, MeOD) δ 9.44 (s, 1H)–hydrogen bonded rotamer H-N_benzimid._, 8.00–7.84 (m, 1H), 7.80 (dd, J = 21.4, 8.3 Hz, 1H), 7.70–7.45 (m, 3H), 7.34 (dd, J = 32.3, 7.9 Hz, 1H), 7.14–7.00 (m, 3H), 4.80 (t, J = 5.4 Hz, 1H), 4.55 (t, J = 5.4 Hz, 1H), 3.97 (t, J = 4.9 Hz, 1H), 3.68 (dd, J = 19.7, 4.9 Hz, 2H), 3.24 (t, J = 6.9 Hz, 1H), 3.03 (dd, J = 15.5, 7.9 Hz, 1H), 2.90 (t, J = 6.8 Hz, 1H); ^13^C NMR (151 MHz, MeOD) δ 156.1, 140.7, 136.7, 130.7, 127.2, 126.9, 126.5, 122.5, 121.0, 118.2, 117.7, 114.3, 112.5, 111.2, 111.0, 46.3, 40.9, 39.4, 24.8.; HPLC-MS analysis t–4.76 min (96% purity, method A), calc. for C_22_H_23_N_9_ *m*/*z* = 413.2, found *m*/*z* = 414.2 [M+H]^+^.

#### 4.1.32. N^2^-(2-(1H-indol-3-yl)ethyl)-N^4^-(2-(2-methyl-1H-benzo[d]imidazol-1-yl)ethyl)-1,3,5-triazine-2,4,6-triamine hydrochloride (**18**)

White solid (55% yield), mp: 190–194 °C; ^1^H NMR (600 MHz, MeOD) δ 7.73–7.63 (m, 1H), 7.63–7.48 (m, 3H), 7.41–7.29 (m, 2H), 7.12 (dd, J = 14.4, 7.8 Hz, 1H), 7.09–7.00 (m, 2H), 4.69 (bs, 1H), 4.44 (t, J = 5.5 Hz, 1H), 3.93 (t, J = 5.1 Hz, 1H), 3.69 (t, J = 4.8 Hz, 1H), 3.62 (t, J = 7.0 Hz, 1H), 3.30 (t, J = 6.9 Hz, 1H), 3.14–2.97 (m, 4H), 2.75 (s, 1H); ^13^C NMR (151 MHz, MeOD) δ 156.1, 150.9, 136.7, 132.0, 127.3, 126.3, 125.9, 122.6, 121.0, 118.3, 117.9, 113.3, 112.1, 111.2, 111.0, 44.3, 41.0, 38.9, 24.7, 10.3; HPLC-MS analysis t–4.25 min (97% purity, method A), calc. for C_23_H_25_N_9_ *m*/*z* = 427.2, found *m*/*z* = 428.2 [M+H]^+^.

#### 4.1.33. N^2^-(2-(1H-Indol-3-yl)ethyl)-N^4^-(2-(2-(Trifluoromethyl)-1H-Benzo[d]imidazol-1-yl)ethyl)-1,3,5-Triazine-2,4,6-Triamine Hydrochloride (**19**)

White solid (76% yield), mp: 108–110 °C; ^1^H NMR (600 MHz, MeOD) δ 8.15–7.72 (m, 2H), 7.61–7.47 (m, 2H), 7.44 (d, J = 7.7 Hz, 1H), 7.35 (d, J = 8.0 Hz, 1H), 7.11 (dd, J = 24.3, 17.3 Hz, 2H), 7.06–6.95 (m, 1H), 3.77 (dt, J = 47.9, 6.7 Hz, 2H), 3.11 (t, J = 6.3 Hz, 2H); ^13^C NMR (151 MHz, DMSO) δ 156.1, 140.8, 136.7, 136.0, 127.5, 125.6, 123.9, 123.3, 121.2, 120.4, 118.6, 118.5, 112.1, 111.8, 111.4, 44.7, 41.1, 30.0, 27.5, 24.9; HPLC-MS analysis t–5.85 min (92 % purity, method A), calc. for C_23_H_22_F_3_N_9_ *m*/*z* = 481.2, found *m*/*z* = 482.2 [M+H]^+^.

#### 4.1.34. General Procedure for the Synthesis of Final Compounds **20**–**22** (Microwave-Assisted)

Briefly, **69** [20] (0.25 g, 0.8 mmol), amines **62**, **66**, and **68** (2.0 mmol) potassium carbonate (0.36 g, 2.5 mmol) and TBAB (0.032 g, 0.1 mmol) were ground in a mortar and transferred to a sealed tube. Subsequently, 5 wt % DMF was added. The mixture was reacted in a microwave reactor at 50 W for 2.5 min. Reaction progress was monitored via TLC (chloroform: MeOH 9:1 *v*/*v*). The mixture was cooled down and extracted with chloroform (3 × 20 mL). Organic layers were combined, dried over MgSO_4_, and concentrated. The crude product was purified via column chromatography with elution using chloroform then chloroform:MeOH (*v*/*v*) 99:1-> 97:3. Colorless sticky oil was then dissolved in acetone, and pH was adjusted to 2–3 with 4 M HCl in 1,4-dioxane. The resulting mixture was crushed by the addition of cold diethyl ether, then the white powder was filtered and rinsed with cold diethyl ether and then dried to yield the final product **20**–**22**.

#### 4.1.35. N^2^-(2-(1H-Indol-3-yl)ethyl)-N^4^-(3-Chlorophenyl)-1,3,5-Triazine-2,4,6-Triamine Hydrochloride (**20**)

White solid (84% yield), mp: 174–178 °C; ^1^H NMR (600 MHz, MeOD) δ 7.80 (d, J = 20.9 Hz, 1H), 7.63–7.42 (m, 2H), 7.38–7.27 (m, 2H), 7.20–7.08 (m, 3H), 7.06–6.98 (m, 1H), 3.84–3.70 (m, 2H), 3.11 (t, J = 7.0 Hz, 2H); ^13^C NMR (151 MHz, MeOD) δ 136.8, 133.9, 129.6, 127.2, 124.0, 122.4, 121.2, 121.0, 119.5, 118.3, 117.7, 111.0, 110.9, 41.6, 24.5; HPLC-MS analysis t–6.53 min (100% purity, method A), calc. for C_19_H_18_^35^ClN_7_ *m*/*z* = 379.1, found *m*/*z* = 380.1 [M+H]^+^, calc. for C_19_H_18_^37^ClN_7_ *m*/*z* = 381.1, found *m*/*z* = 382.1 [M+H]^+^.

#### 4.1.36. N^2^-(2-(1H-Indol-3-yl)ethyl)-N^4^-(4-Fluorophenyl)-1,3,5-Triazine-2,4,6-Triamine Hydrochloride (**21**)

White solid (82% yield), mp: 232–235 °C; ^1^H NMR (600 MHz, MeOD) δ 7.69–7.46 (m, 3H), 7.36 (d, J = 8.1 Hz, 1H), 7.16–6.99 (m, 5H), 3.75 (bs, 2H), 3.09 (bs, 2H); ^13^C NMR (151 MHz, MeOD) δ 136.8, 127.2, 123.8, 122.3, 121.0, 118.3, 117.7, 115.0, 114.8, 111.2, 110.9, 41.4, 24.6; HPLC-MS analysis t–5.97 min (100% purity, method A), calc. for C_19_H_18_FN_7_ *m*/*z* = 363.2, found *m*/*z* = 364.4 [M+H]^+^.

#### 4.1.37. N^2^-(2-(1H-Indol-3-yl)ethyl)-N^4^-(4-Methoxyphenyl)-1,3,5-Triazine-2,4,6-Triamine Hydrochloride (**22**)

White solid (83% yield), mp: 194–196 °C; ^1^H NMR (600 MHz, MeOD) δ 7.61–7.38 (m, 3H), 7.36 (d, J = 8.1 Hz, 1H), 7.15–7.08 (m, 2H), 7.06–6.91 (m, 2H), 6.84 (bs, 1H), 3.84–3.67 (m, 5H), 3.07 (bs, 2H); ^13^C NMR (151 MHz, MeOD) δ 136.8, 127.2, 123.5, 122.3, 121.0, 118.3, 117.8, 114.1, 113.6, 111.2, 110.9, 54.5, 41.4, 24.6; HPLC-MS analysis t–5.88 min (100% purity, method A), calc. for C_20_H_21_N_7_O *m*/*z* = 375.2, found *m*/*z* = 376.1 [M+H]^+^.

### 4.2. Radioligand Assay

The cell culture, cell membranes, and radioligand binding assays were performed in accordance with standard protocols [17].

### 4.3. Atlas Activity

Activity maps were prepared using the Atlas Activity tool available in Flare [24] according to the procedure described in a previous paper [15].

### 4.4. Molecular Modelling

A homologous model developed by us earlier was used for molecular modeling studies [15]. The docking procedure and the QM–MM optimization of the ligand–receptor complex were performed according to a previously described protocol [15,20,32].

### 4.5. Metabolic Stability

All assays were performed according to the protocols described previously [15,33,34,35].

### 4.6. CYP3A4 Activity

All assays were performed according to the protocols described previously [15,33,34,35].

### 4.7. Hepatotoxicity

All assays were performed according to the protocols described previously [15,33,34,35].

### 4.8. In Vivo Cardiotoxicity

To determine the toxicity of the compounds, the fish embryo toxicity (FET) test was performed on zebrafish (Danio rerio) according to OECD Test Guideline 236. The collected embryos were transferred to a Petri dish with E3 medium (5 mM NaCl, 0.33 mM MgCl_2_, 0.33 mM CaCl_2_, 0.17 mM KCl; pH 7.2) and then placed in 6-well plates, 10 embryos per well. Stock solutions **2** and **12** were prepared in DMSO. In these experiments, the range of different concentrations of the solutions was prepared by dissolving stock solutions in the E3 medium each time directly before addition to the wells. The solutions were changed once daily, and the embryos were maintained in the incubator at 28.5 °C. At the end of the exposure period (96 hpf–hours postfertilization), acute toxicity was determined based on a positive outcome in any of the four visual indicators of lethality, including the coagulation of fertilized eggs, lack of somite formation, lack of detachment of the tailbud from the yolk sac and lack of heartbeat. The value of LD_50_ was calculated. Heartbeats were recorded to observe cardiotoxic effects. Moreover, images of the fish from each group were taken at the final time point to monitor the occurrence of developmental malformations. For the observations, a Discovery V8 Stereo optical microscope and Zeiss hardware were used. A dose-response curve was generated using Prism 8.0.1 (GraphPad Software) by fitting a sigmoid curve model to experimental data points. The concentrations of the compounds of interest causing 50% mortality (LD_50_) of 96 hpf larvae of Danio rerio were calculated.

## Data Availability

Not applicable.

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
