# Peer review of "Design, Synthesis and Biological Evaluation of Novel 1,3,5-Triazines: Effect of Aromatic Ring Decoration on Affinity to 5-HT7 Receptor"

_ijms, 2022, doi:10.3390/ijms232113308_

Round 1

Reviewer 1 Report

Some recent studies have begun to explore in detail the interaction of 1,3,5 triazines with different serotonin receptors.  For example, a potential lead compound has been identified with high affinity binding to the 5-HT6 receptor that also shows anxiolytic activity and improves performance in cognitive tasks.  In this paper, the authors use a similar approach to determine whether such an approach would identify triazine derivatives that would be selective ligands for the 5-HT7 receptor.  To this end, the authors have extended their prior studies of a novel tryptamine core connected with an aminotriazine to produce novel compounds with selective binding to 5-HT7. 

Strengths: 

1.      The rationale for developing novel compounds in this class is strong

2.      The authors successfully identified 5’ indole derivatives with high binding affinity to 5HT-7 but low/no binding to a variety of other 5-HT receptors.

Weaknesses:

1.      Significant cytotoxicity in both cultured cells and zebrafish embryos at uM concentrations, especially leading to heart defects.  These findings indicate these new compounds are not biologically useful at present but modifications could minimize cytotoxic effects.

Overall:

There is real value to producing and assessing novel structures that show selective binding to the 5-HT receptor but lack the arylpiperazine pharmachore.   The finding that these novel triazines, despite cytotoxic effects, exhibited specific 5-HT7 binding highlights the fact that additional modifications of these compounds may minimize side effects while retaining specific binding.

Author Response

I would like to thank for all remarks. We are currently working on safety aspects for considered compounds.

Reviewer 2 Report

1.     Please add the SD values for Ki in Table 1.

2.     Figure 3, it is not easy to tell the difference between 3A and 3B. Please only keep the most representative structures for side-by-side comparison. Also please increase the image resolution as well.

3.     Figure 4, similar recommendations as for Figure 3. Also please display the models in multi-angles for better clarification.

4.     Line 249, what’s the meaning of ‘TMh’, please add a footnote for it.

5.     Line 317, please add the reference for OECD 236 test.

Author Response

I would like to thank for all remarks.

Ad 1. SD has been added

Ad 2. I have changed figure 3 A a B to make them more easy to analyse. I add only 2 represeantative active molecules as well as 3 representative inactive compounds

Ad 3. As for Ad 2

Ad 4. An explanation has been added

Ad 5. Reference has been added